# Controlling gain one photon at a time

**Gregory W Schwartz[1,2], Fred Rieke[1,2]\***

[1]Department of Physiology and Biophysics, University of Washington, Seattle, United States; [2]Howard Hughes Medical Institute, University of Washington, Seattle, United States

**Abstract** Adaptation is a salient property of sensory processing. All adaptational or gain control mechanisms face the challenge of obtaining a reliable estimate of the property of the input to be adapted to and obtaining this estimate sufficiently rapidly to be useful. Here, we explore how the primate retina balances the need to change gain rapidly and reliably when photons arrive rarely at individual rod photoreceptors. We find that the weakest backgrounds that decrease the gain of the retinal output signals are similar to those that increase human behavioral threshold, and identify a novel site of gain control in the retinal circuitry. Thus, surprisingly, the gain of retinal signals begins to decrease essentially as soon as background lights are detectable; under these conditions, gain control does not rely on a highly averaged estimate of the photon count, but instead signals from individual photon absorptions trigger changes in gain.

## Introduction

Sensory systems encode an enormous range of input signals—for example, a rock concert is $\sim 10^{12}$ times louder than a just detectable whisper. Maintaining sensitivity as inputs change requires adaptational mechanisms that adjust the gain or amplification of neural signals to match their limited dynamic range to the range of the input signals a cell receives. Such gain control mechanisms operate under challenging conditions as the inputs they encounter can vary rapidly in time and locally in the space of possible stimuli. For example, visual neurons can experience >1000-fold changes in input a few times a second as the eyes move to explore a typical scene (*Frazor and Geisler, 2006*). Standard cameras fail to capture the full structure of such scenes because they employ a single global gain control in the form of an exposure setting, which is often poorly matched to the brightest and dimmest regions of a scene. Sensory gain controls are clearly more sophisticated.

Effective gain control requires balancing the need to be rapid and local with the need to be accurate (reviewed by *Rieke and Rudd, 2009*). This is an example of the classic change detection problem: how many samples from a signal distribution are needed to determine whether or not the distribution has changed (*Buracas et al., 1998*; *Deneve, 2008*; *Wark et al., 2009*)? Gain control mechanisms operating near absolute visual threshold exemplify this problem. Human behavioral threshold begins to increase for backgrounds producing $\sim 0.01$ isomerized rhodopsin molecules per rod per second (R\*/rod/s) (reviewed by *Donner, 1992*). Under these conditions, the number of photons absorbed in a small time window fluctuates widely due to the Poisson statistics that govern photon absorption. In principle, increases in behavioral threshold could solely reflect this increased noise without gain changes (*Barlow, 1965*). However, several studies suggest that weak backgrounds decrease the gain of retinal signals (*Donner, 1992*; *Brown and Rudd, 1998*). Retinal gain control mechanisms operating at these low light levels must work in the presence of the irreducible noise associated with random photon arrivals.

At low light levels, rod-mediated signals traverse the retina through the specialized rod bipolar pathway: rods → rod bipolar cells → AII amacrine cells → cone bipolar cells → ganglion cells (*Figure 1A*; for review see *Bloomfield and Dacheux, 2001*; *Field et al., 2005*). The total number of rods that convey the signal increases at each stage of this pathway (*Figure 1B*). This convergence, together with

**\*For correspondence:**
rieke@u.washington.edu

**Competing interests:** The authors declare that no competing interests exist.

**Reviewing editor**: Ronald L Calabrese, Emory University, United States

**eLife digest** To process the sights and sounds around us, our senses must be attuned to a huge range of signals: from barely audible whispers to deafening rock concerts, and from dim glimmers of light to bright spotlights. Sensory neurons face the challenge of encoding this huge range of inputs within their much more restricted response range. Thus, neurons in our eyes and ears must continually adjust their gain or sensitivity to match changes in the light and sound inputs. These gain control processes must operate rapidly to keep up with the ever-changing input signals, but must also operate accurately so as not to distort the inputs.

The trade-off between rapid and accurate gain control can be illustrated by considering how the retina processes information at low light levels. There are two main types of light-sensitive cells in the retina: rods and cones. Vision at night relies on the ability of the rods to detect single photons—the smallest unit of light. In starlight, an individual rod will register photons only rarely, and most of the time, the majority of the rods will not register any photons. Neurons in the retinal circuits that read out the rod signals receive input from hundreds or thousands of rods, and those rod inputs are highly amplified to allow detection of the responses produced when a tiny fraction of the rods absorbs a photon. But this amplification is dangerous, as it could easily saturate retinal signals when light levels increase. Gain control mechanisms are needed to avoid such saturation.

Schwartz and Rieke now add to our understanding of this process by examining how the retinas of non-human primates behave in low light. They reveal that levels of background light that can only just be detected behaviorally trigger retinal gain controls; these gain controls operate when less than 1% of rods absorb a photon. Under these conditions, the physics of light itself will cause considerable variability in the stream of photons arriving at the retina, leading to high variability in the gain of retinal responses. Nonetheless, changes in gain occurred rapidly following changes in background, indicating that the underlying mechanisms spend little time averaging incident photons. Taken together, these findings will require revisiting our ideas about how adaptational mechanisms balance the competing demands of speed and reliability to help us see the world around us.

amplification of signals by cellular and synaptic mechanisms, permits fully dark-adapted ganglion cells to generate one or more spikes to flashes producing photon absorptions in only ~0.1% of the rods (*Barlow et al., 1971*; *Mastronarde, 1983*; P Ala-Laurila and F Rieke, unpublished). Left unchecked, this high gain would cause just detectable backgrounds to produce ganglion cell firing rates in excess of 100 spikes per second. Gain control mechanisms are needed to prevent such excessive firing.

Mechanisms controlling the gain of rod-mediated signals have been identified within the phototransduction cascade in the rods themselves (*Schneeweis and Schnapf, 2000*; *Dunn and Rieke, 2006*), and at the synapse between rod bipolar cells and AII amacrine cells (*Dunn and Rieke, 2008*). But the greater convergence at downstream locations in the rod bipolar pathway could provide a more reliable signal to control gain. Hence, we explored how the gain of retinal signals is controlled in dim backgrounds, with three main goals: (1) to determine if the gain of retinal signals is altered by backgrounds near the onset of changes in human behavioral threshold, (2) to determine how the retinal circuit averages over space and time to detect changes in the photon flux, and (3) to identify novel sites of gain control in the rod bipolar pathway.

## Results

We focus on responses of primate On parasol (magnocellular projecting) ganglion cells. On parasol ganglion cells provide information to cortical circuits about subtle changes in contrast (*Kaplan and Shapley, 1982*) and receive abundant rod-mediated input at low light levels (*Purpura et al., 1988*; *Field et al., 2009*). Thus, these cells are likely to contribute critically to vision at low light levels. On parasol cells also provide a sensitive measure of the output of the rod bipolar pathway. This pathway dominates rod-mediated signaling across ganglion cell types at low light levels, and hence should provide shared steps in processing of rod signals. Indeed, as described below, we find that the gain of retinal signals at low light levels is controlled at a site located within the rod bipolar pathway.

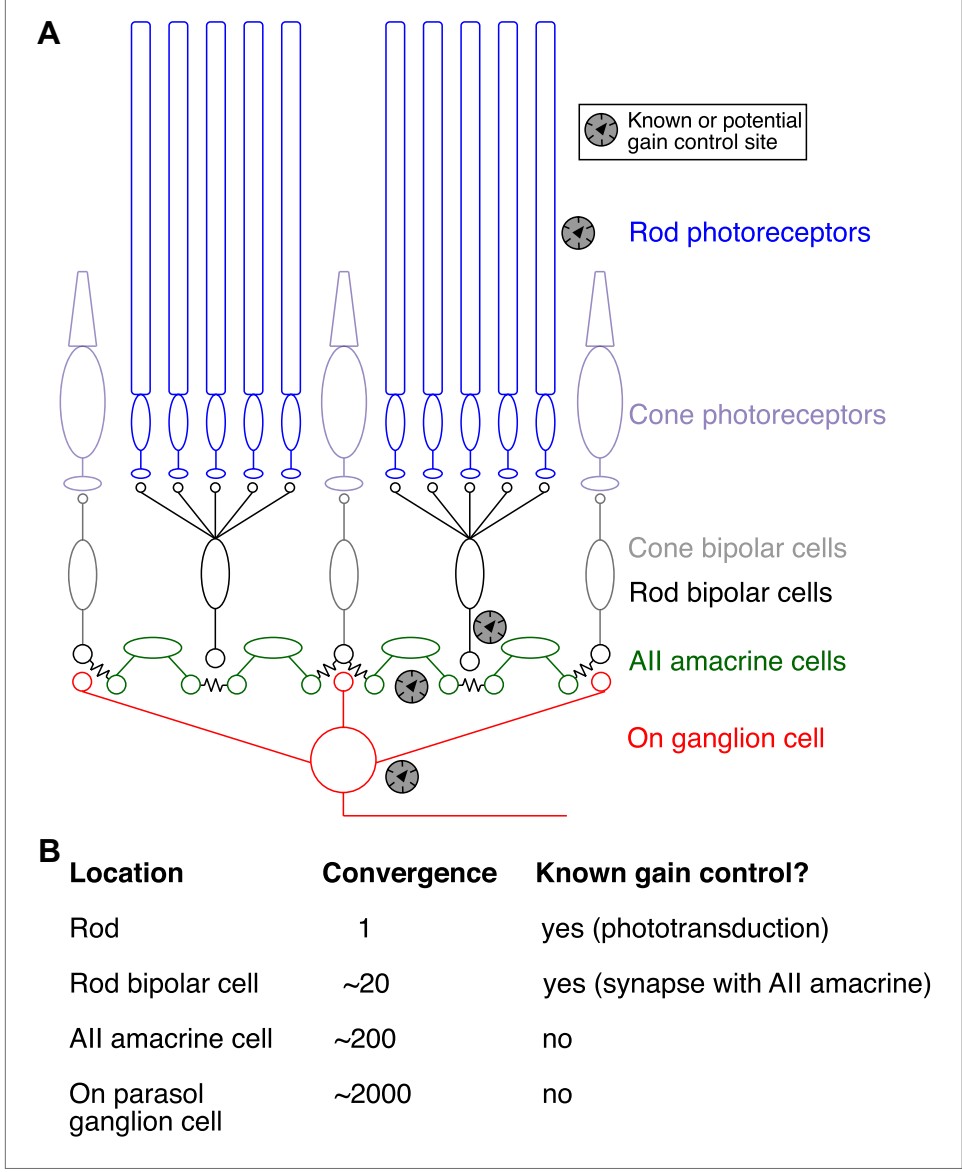

**Figure 1**. Gain control in the rod pathway. (**A**) Simplified diagram of the excitatory circuits that relay rod and cone signals to On ganglion cells. Elements exclusive to the cone pathway are shown in light colors. Chemical synapses are excitatory and are indicated by apposition of small circles or ellipses; chemical synapses in the rod bipolar circuit are found between rods and rod bipolar cells, between rod bipolar cells and AII amacrine cells, and between cone bipolar cells and ganglion cells. Electrical synapses are indicated by connected cells—for example, between cone bipolar cells and AII amacrine cells and between neighboring AII amacrine cells. 'Gain knob' icons indicate known or potential sites of gain control (see 'Introduction' for details). (**B**) Table of convergence (number of rods providing input) and previously described gain control mechanisms at each circuit location. Convergence numbers are estimated for an On parasol ganglion cell in peripheral macaque retina.

## Gain controls in the retinal circuitry operate near the onset of behavioral threshold changes

We measured spike responses (*Figure 2A*) and excitatory synaptic inputs (*Figure 2B*) of On parasol ganglion cells in response to brief flashes superimposed on a range of backgrounds (see also 'Materials and methods' and *Figures 10 and 11*). We focused specifically on backgrounds near the onset of changes in human behavioral threshold (0.01–0.2 isomerizations/rod/s or R*/rod/s). Flash strengths (0.001–0.006 R*/rod) were chosen to elicit responses well below saturation. Both individual responses

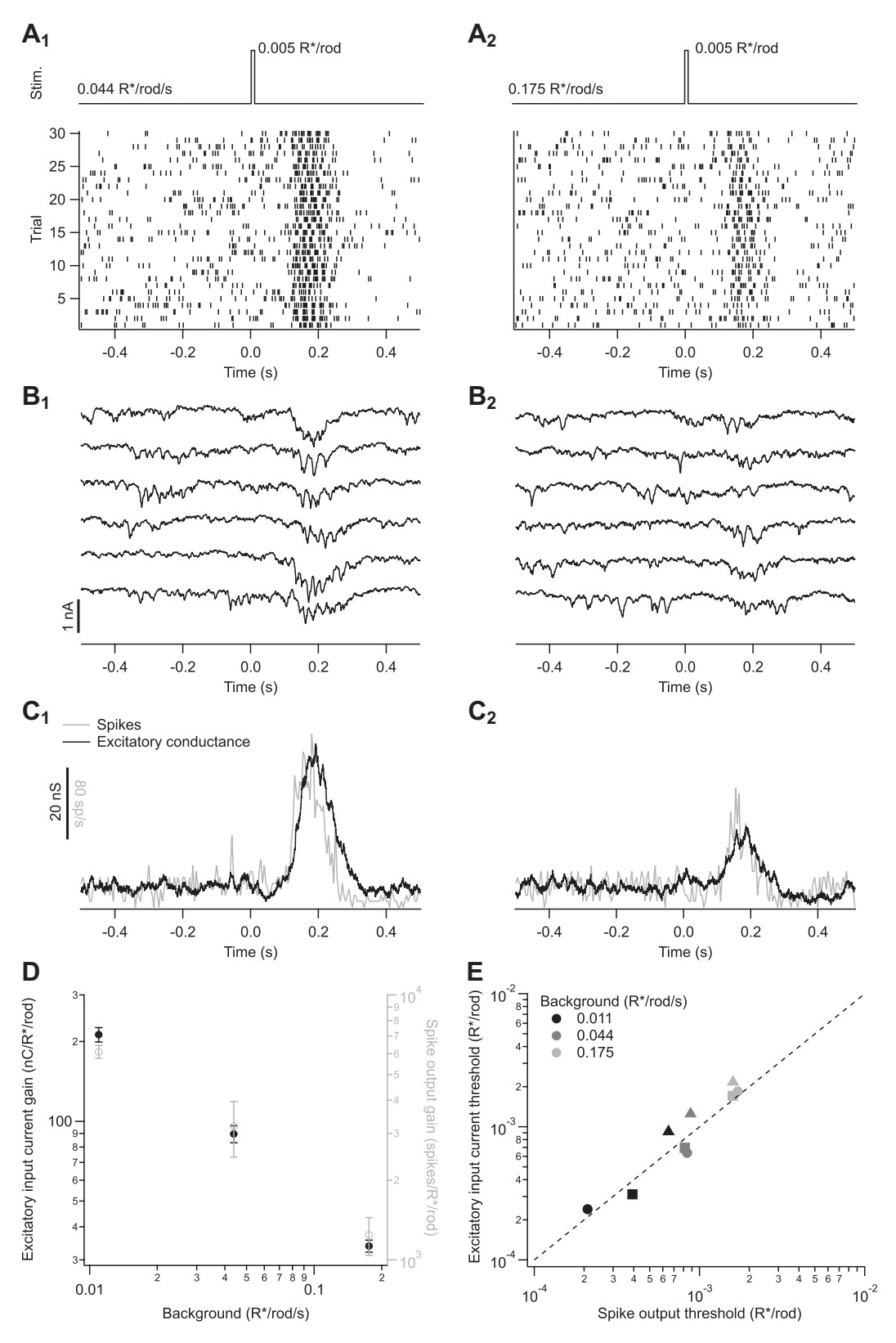

**Figure 2**. Adaptation is engaged at dim backgrounds in both excitatory inputs and spike outputs from On parasol cells. (**A**)–(**C**) responses from an individual On parasol ganglion cell to dim flashes at two different backgrounds. (**A₁**) Light stimulus (top, 20 ms flashes at 0 s) and raster of the cell's spike
*Figure 2. Continued on next page*

*Figure 2. Continued*

response to several stimulus trials. (**B₁**) Excitatory synaptic currents in response to the same stimulus. (**C₁**) Mean spike rate and excitatory conductance. (**A₂**)–(**C₂**) responses to the same flash as in (**A₁**–**C₁**) but delivered on a higher background. (**D**) Gain of the excitatory input current and spike output as a function of background for the cell in (A–C). Gain of the excitatory currents was measured by integrating the current over time and dividing by the flash strength, and hence has units of nC/R*/rod/s. Gain of the spike responses was measured by integrating spikes over time. Error bars are SD (*n* = 20–30 trials). (**E**) Detection threshold (see 'Materials and methods') calculated from excitatory input current and spike output for three different cells (symbols) at three backgrounds each (gray scale). Dashed line represents unity.

(*Figure 2A,B*) and averages (*Figure 2C*) had a clear dependence on background. Furthermore, a cell's excitatory synaptic input and its spike response changed similarly with changes in background (*Figure 2C*). To quantify the effect of background on a cell's response, we determined the response gain by dividing the response amplitude (spike count or integrated current) by the flash strength; the gain measures the response per R*/rod. The gain of spike responses and excitatory input had a similar dependence on background (*Figure 2D*).

*Figure 2C,D* suggest that the effect of changes in background on a cell's spike output is dominated by the changes in excitatory input. To further test this conclusion, we defined a detection threshold as the strength of the flash (in R*/rod) eliciting a response with a signal-to-noise ratio of 1 (see 'Materials and methods'). Noise was measured from sections of recording prior to the flash. Detection thresholds for spike responses and excitatory synaptic inputs measured in the same cell were near identical across a range of backgrounds (*Figure 2E*). Consistent with the similarity of the spike responses and excitatory inputs, inhibitory synaptic inputs to the same stimuli were three to four times smaller than the excitatory inputs, and hence unlikely to contribute strongly to spike output (data not shown).

*Figure 2* indicates the changes in the gain of an On parasol cell's spike output are dominated by changes in the cell's excitatory synaptic inputs. Below, we focus on how gain of the excitatory synaptic inputs is controlled.

## Dependence of gain and noise on background

Changes in human behavioral threshold with background show several characteristic regions (reviewed by *Barlow, 1965*; *Donner, 1992*; *Rieke and Rudd, 2009*): (1) a 'dark light' region extending to ~0.01 R*/rod/s in which threshold is independent of background, (2) a 'Rose–deVries' region in which threshold increases with the square root of the background (*de Vries, 1943*; *Rose, 1948*), and (3) a 'Weber' region in which threshold increases proportionally with the background. Since threshold depends on both signal and noise, the relationship between these behavioral measurements and gain control is unclear. Hence, the experiments below characterize gain, noise, and threshold of the On parasol responses across a range of background intensities spanning these behaviorally defined regions.

We extended experiments like those in *Figure 2* to span a wide range of backgrounds (*Figure 3*). In this and all subsequent experiments, we adjusted the flash strength to elicit similar responses across backgrounds because the change in gain exceeded the cell's linear response range (see 'Materials and methods' and *Figures 10 and 11*). Adjusting flash strength (and dividing the response by the flash strength to measure gain) allowed us to probe a ~10,000-fold range of backgrounds. Across the entire range tested, the dependence of gain on background (*Figure 3A*) was well described by a modified Weber function:

$$G = \frac{G_D}{1 + I_B/I_0},$$

where G is the gain and $G_D$ is its maximum value, $I_B$ is the background, and $I_0$ is the background required to reduce the gain by 50%. Gain in Equation 1 is constant when $I_B \ll I_0$, begins to decrease when $I_B$ nears $I_0$, and scales inversely with background when $I_B \gg I_0$. For the cell in *Figure 3A*, $I_0$ was 0.076 R*/rod/s (marked by the arrowhead); across cells, $I_0$ was 0.08 ± 0.03 R*/rod/s (mean ± SD, n = 6). We observed a small but consistent reduction in gain in darkness relative to its maximal value at backgrounds ~0.01 R*/rod/s. The reduced gain in darkness was highly dependent on the history of light exposure, and hence is only partially captured in *Figure 3A* because we returned to the dark condition after exposure to several backgrounds (see 'Recordings and light stimulation'). The low dark gain reflects a nonlinear processing step that is engaged in the dark but relieved for backgrounds

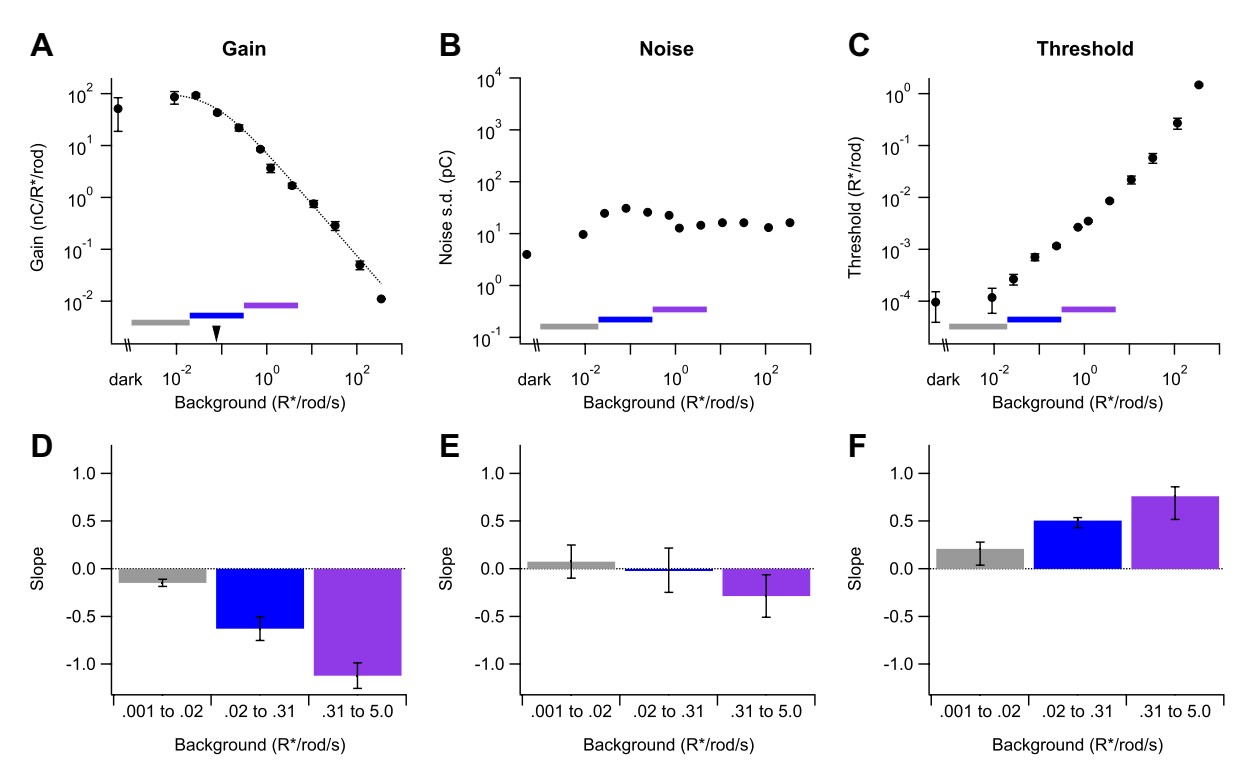

**Figure 3**. Changes in gain, noise, and threshold with background. (**A**)–(**C**) Example data from a single cell showing gain of the dim flash response (**A**), noise (**B**), and threshold (**C**) as a function of background (see 'Materials and methods'). Error bars (**A** and **C**) are standard deviations across two flash strengths. Colored bars indicate background segments in which the slope was computed. Dotted line (**A**) represents the fit of a modified Weber function with the half-desensitizing background indicated by an arrowhead (see Equation 1). (**D**)–(**F**) Population data reporting the slope of each parameter on a log–log scale in three background regions. Error bars are SD ($n$ = 4, 6, and 5 cells in each background segment).

producing 0.005–0.01 R*/rod/s; this nonlinearity and its impact on visual sensitivity is the subject of another study (P Ala-Laurila and F Rieke, unpublished).

Two features of *Figure 3A* are notable. First, over a ~1000-fold range of backgrounds gain closely followed Weber law behavior, even though multiple mechanisms contribute to gain control across this range. This indicates an orderly coordination of gain control mechanisms (see 'Discussion'). Second, gain began to decrease for backgrounds near 0.01 or 0.02 R*/rod/s—close to the onset of changes in human behavioral threshold. We summarized the onset of gain changes across cells by choosing three regions spanning equal ranges of background intensities centered below, on, or above $I_0$ (horizontal colored lines in *Figure 3A*). *Figure 3D* plots the exponent describing the dependence of gain on background (i.e., the slope on a log–log scale as in *Figure 3A*) in each region. Gain scaled inversely with background above 0.3 R*/rod/s, and more gradually at lower background intensities. Consistent with the ability of Equation 1 to describe the data, we did not see evidence for an extended region where gain scaled inversely with the square root of the background (see 'Discussion').

We quantified noise at each background by projecting the current prior to the flash along the same template used to measure signal (see 'Analysis'). Noise also changed with background, but such changes were modest across those backgrounds near the onset of changes in behavioral threshold (*Figure 3B,E*); specifically, noise changed <10-fold over a range of backgrounds that produced an ~10,000-fold change in gain. These changes in noise with background were much less than expected from Poisson fluctuations in photon absorption and no changes in gain. If Poisson fluctuations dominated noise and gain remained constant, the standard deviation of the current should increase as the square root of the background—that is, with slope 0.5—since the variance of a Poisson process is proportional to the mean. Noise changed much less with background than this prediction (*Figure 3B,E*). The largest increases in noise occurred between darkness and backgrounds <0.01 R*/rod/s.

These changes are produced by a nonlinear processing step that reduces gain in the dark (see above; P Ala-Laurila and F Rieke, unpublished). The reduction in noise at backgrounds higher than 0.1 R*/rod/s likely reflects the stronger decrease in gain at these backgrounds, which more than compensates the increase in Poisson fluctuations.

Changes in noise, although modest, had a noticeable effect on detection threshold—defined as the strength of the flash eliciting a response with a signal-to-noise ratio of 1 (*Figure 3C,F*; see 'Analysis'). This measure of threshold assumes a linear scaling of response with flash strength, which fails in the dark but holds for backgrounds exceeding 0.01 R*/rod/s (P Ala-Laurila and F Rieke, preprint). The threshold in *Figure 3C,F* corresponds to the threshold of an ideal observer of the ganglion cell's output, assuming that signal and noise are independent and additive. Thus, the threshold defined here provides a point of comparison to human behavioral threshold vs intensity curves. The threshold for On parasol responses increased for backgrounds as low as 0.02 R*/rod/s, in close correspondence with backgrounds that produce changes in behavioral threshold (*Donner, 1992*). Threshold increased with a slope <1 for backgrounds between 0.02 and 5 R*/rod/s (*Figure 3F*). The extended region over which the slope was less than unity reflected the increase in noise at backgrounds lower than those required to produce gain changes, a region near the onset of gain changes where noise was near constant, and the decrease in noise at backgrounds where gain began to decrease proportionally with background.

## Properties of gain control

The experiments described above indicate that the gain of retinal signals begins to decline at backgrounds producing ~0.02 R*/rod/s. Gain control under these conditions is challenging because photons arrive rarely at individual rod photoreceptors; retinal responses at these backgrounds are noisy, and hence changes in background will be difficult to discern without a large degree of spatial and/or temporal averaging. At the same time, left unabated, the high gain associated with fully dark-adapted vision would cause a parasol ganglion cell to generate hundreds of spikes per second for backgrounds producing ~0.02 R*/rod/s. The experiments described in this section characterize several key properties of how gain is controlled—particularly the statistical feature of the input that sets gain and how the mechanism integrates over space and time.

## Gain is controlled by mean not fluctuations

Changes in the mean background necessarily also change the magnitude of the fluctuations about the mean because of the division of light into discrete photons and the resulting statistical fluctuations in photon arrival. Such Poisson fluctuations in photon absorption are particularly large relative to the mean absorption rate at low backgrounds. Because these statistical fluctuations increase with the square root of the background, a gain control that directly depends on the magnitude of the noise would cause gain to scale inversely with the square root of the background (*Brown and Rudd, 1998*). The experiments below, however, indicate that gain at low backgrounds is controlled by the mean photon absorption rate and not by the magnitude of the fluctuations about the mean rate.

To isolate the potential effects of fluctuations about the mean on gain, we delivered a series of flashes before, during, and after a high contrast fluctuating stimulus while maintaining a constant mean background (*Figure 4A*). A Rose–deVries gain control mechanism based on the fluctuations about the mean predicts that gain should decrease linearly with standard deviation (dashed line in *Figure 4B*). Instead, contrast stimuli that substantially increased the variance of a ganglion cell's excitatory synaptic inputs tended to increase, rather than decrease, the gain of the flash response. In nine cells, stimuli that increased the standard deviation of the baseline excitatory synaptic inputs by 43 ± 17% (mean ± SD) produced a 18 ± 11% increase in gain (*Figure 4B*). These data argue directly against noise controlling gain at low backgrounds, indicating instead that gain is controlled by the mean photon flux.

## Gain changes rapidly

Gain control at low backgrounds requires averaging across rods and/or time to obtain a sufficient photon count to determine that the mean background has changed (*Rushton, 1965*). Too much spatial or temporal averaging could limit the effectiveness of gain control: spatial averaging across regions of a scene with different backgrounds could fail to protect retinal signals from local saturation, and temporal averaging could be ineffective if it introduces too long a delay between changes in input and changes in gain. What strategy, then, does the circuit adopt? We start by describing the kinetics of gain changes.

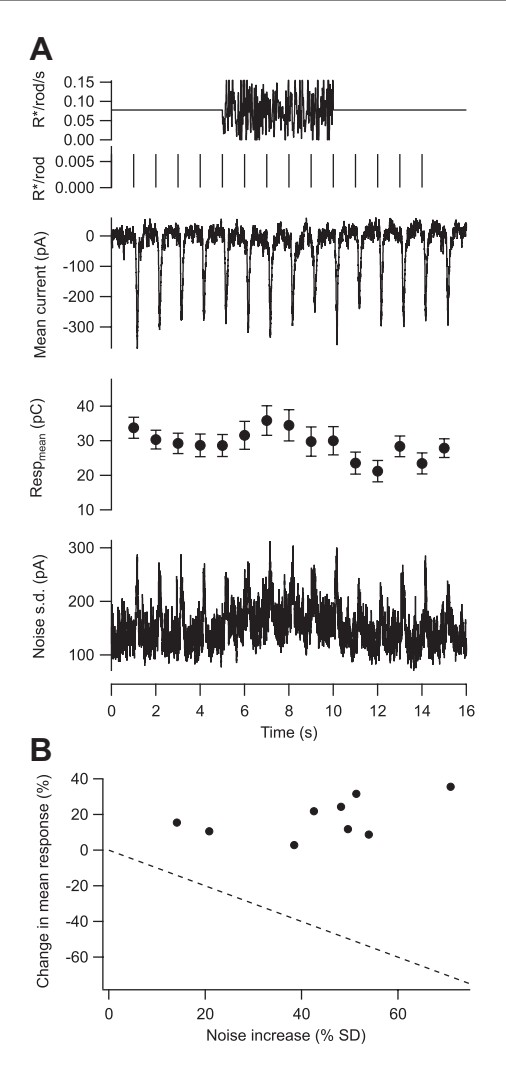

**Figure 4**. Gain is controlled by the mean not the variance of the background. (**A**) Fluctuating background experiment for an example cell. From top: background, probe flashes, mean excitatory input current, mean and SEM of each flash response (*n* = 10 trials), and standard deviation of the current. (**B**) Change in the mean of the flash response plotted against the change in the standard deviation of the baseline current caused by the fluctuating background for nine cells. Dashed line indicates the prediction for a gain control mechanism based on the noise of the background—that is, if gain was inversely proportional to the standard deviation.

To test for slow changes in gain that would reflect extensive temporal averaging, we switched the background periodically between two levels while delivering flashes every second to monitor gain. The baseline firing rate and mean excitatory input changed slowly following a change in background, and had not reached an obvious steady-state level even after 200 s. Gain changes, however, occurred within a few seconds following a change in background—for example, in *Figure 5—figure supplement 1*, the ~twofold change in gain between panels B and C is largely complete by the time of the first flash response (1 s after the background change). Gain did not change noticeably in the subsequent 60–200 s (*Figure 5—figure supplement 1*; see also *Figures 10 and 11*).

To resolve the initial rapid change in gain, we delivered flashes at times ranging from 0.1 to 2 s following either an increase or a decrease in background while measuring a cell's excitatory synaptic inputs. We adjusted the flash strength to generate a near-constant response at the two backgrounds. Subtracting the response to the background step alone isolated responses to the flashes (*Figure 5A*, bottom). Following an increase in background, gain fell with a time constant of 0.10 ± 0.02 s (mean ± SEM, *n* = 5); following a decrease in background, gain increased with a time constant of 0.4 ± 0.1 s (*n* = 6). The decrease in gain at light onset tracked the development of the step response, with apparently little delay due to the gain control mechanism itself. Recovery of gain following a decrease in background took more time, as indicated by the slower kinetics at light offset.

## The gain control mechanism has access to signals from many rods

The rapid kinetics of gain changes at both light onset and offset indicate a need for considerable integration of signals across rods. For example, in the 0.1 s following an increase in background from 0.02 to 0.08 R*/rod/s, <1 in 100 rods absorbs a photon. Thus, the observed gain changes cannot be explained by a mechanism operating on a small collection of rods; for example, a mechanism operating in single rods would only be engaged in 1 out of 100 rods, and hence could reduce overall gain at most by 1%. The halving of gain produced by this step in background requires a mechanism that integrates over at least 50 rods even if a single absorbed photon reduces gain to 0 (*Figure 6*). In this section, we describe experiments that probed the spatial scale of the gain control mechanism by measuring the extent to which a decrease in gain in one region of space transferred to neighboring regions.

We adapted subregions within the ganglion cell receptive field while probing the gain of responses elicited at distinct locations (*Figure 7A*). In the 'baseline' phase of the experiment, we flashed two different probe patterns consisting of a set of bars on a weak spatially uniform adapting background

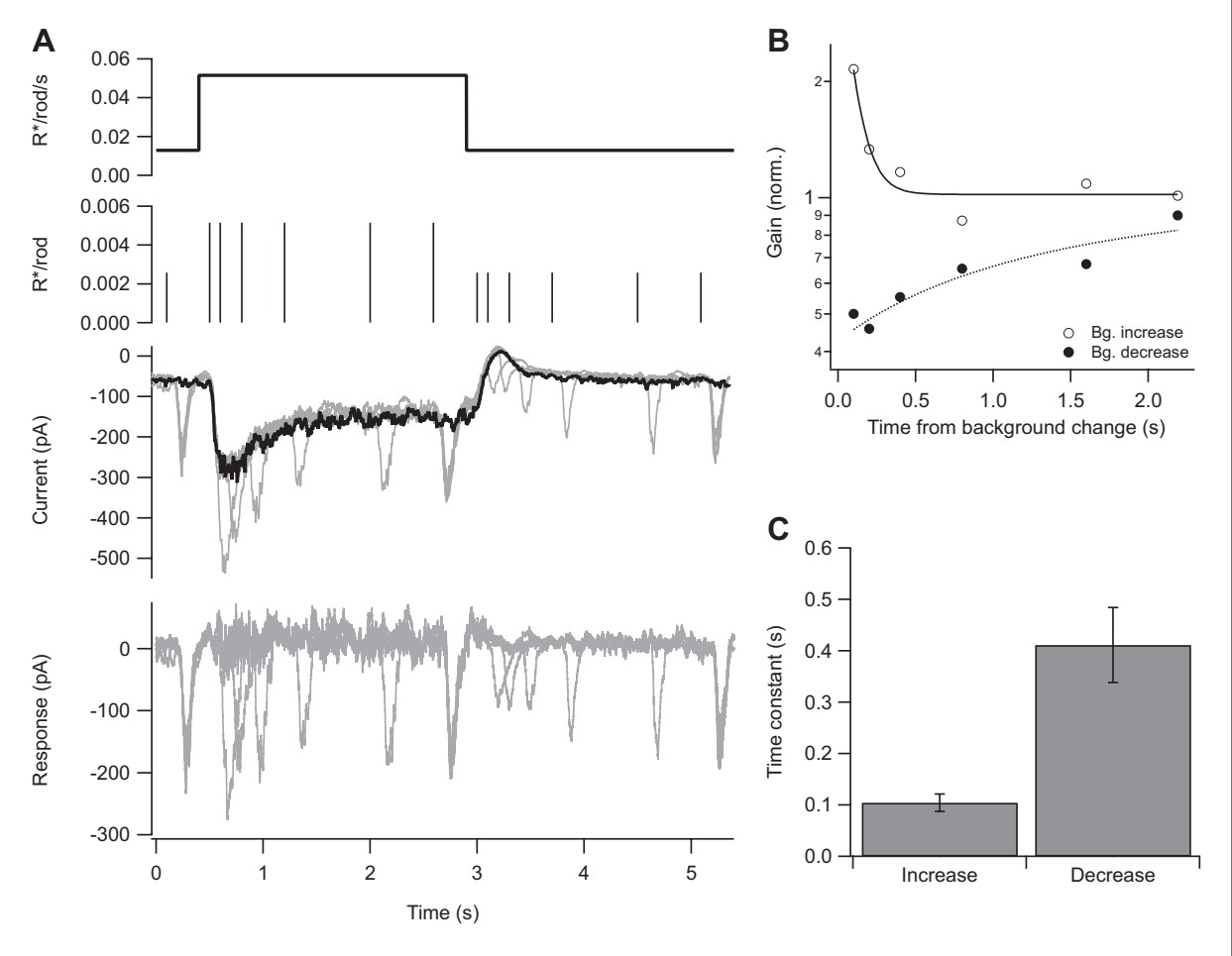

**Figure 5**. Kinetics of gain control. (**A**) From top: background, probe flashes, current traces for each probe flash set (see 'Materials and methods') and for the background alone (black), response obtained by subtracting the background step trace from each test response. Five test flashes where given in each epoch: two at variable times following background step onset and offset, and three at fixed times before the step, at the end of the step, and at the end of the epoch. (**B**) Gain change (on a logarithmic scale) as a function of the time of the probe flash following an increase or decrease in background. Lines are single exponential fits. (**C**) Time constant of the fit as in (**B**) across a population of cells. Error bars are SEM (*n* = 5).

The following figure supplements are available for figure 5:

**Figure supplement 1**. Kinetics of changes in baseline and gain differ.

(0.02 R*/rod/s). In the 'adaptation' phase of the experiment, we presented a steady adapting bar pattern (bar intensity = 0.2 R*/rod/s) that overlapped exactly with one of the probe patterns ('adapted' location) and had no overlap with the other probe pattern ('unadapted' location). We then repeated measurements of the responses to the two probe patterns to determine the extent to which the adapting pattern affected gain in the adapted and unadapted locations.

By exploring a range of widths for the adapting bars, we were able to measure the distance over which gain changes from the adapted location transferred to the unadapted location. *Figure 7B* illustrates the logic of the experiment. Adapting bars that are narrow compared to the adapting subregions (circles in *Figure 7B*) should produce near-identical gain changes in each subregion since the total adapting pattern will vary little between subregions. In this case, the ganglion cell should respond similarly to probes at the adapted and unadapted locations, indicating complete or near-complete transfer of gain changes. Adapting bars that are wide compared to the size of the adapting subregions should produce variable changes in gain across subregions; in this case, the ganglion cell should respond more strongly at the unadapted location, indicating incomplete transfer of gain changes.

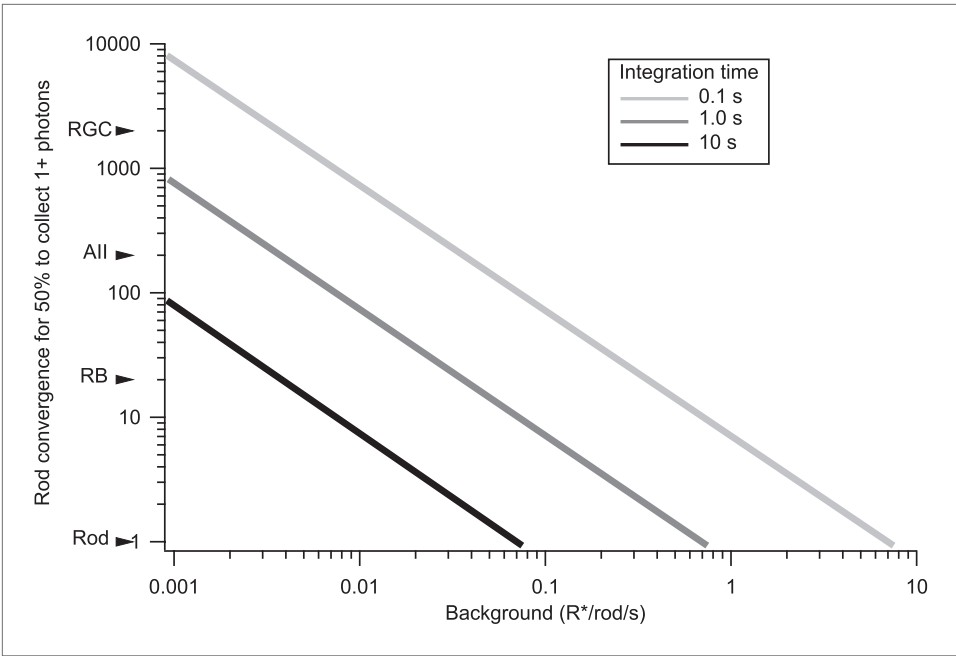

**Figure 6**. The trade-off between rod convergence and integration time in gain control. Rod convergence required for 50% of the neural elements to collect one or more photons is plotted as a function of background (see 'Materials and methods'). Estimated convergence along the rod bipolar pathway is indicated along the y-axis. AII: AII amacrine cell; CB: On cone bipolar cell; RB: rod bipolar cell; RGC: peripheral parasol retinal ganglion cell.

*Figure 8A* shows results from one such experiment. Gain changes at the adapted and unadapted location were measured by dividing responses to probes during the adaptation phase (gray traces) by those during the baseline phase (black traces). Adaptation transfer was defined as the gain change at the unadapted location divided by the gain change at the adapted location. For the cell in *Figure 8A*, the narrow bars produced considerably greater adaptation transfer than the wide bars.

*Figure 8B* collects results across bar sizes and cells. Gain changes transferred completely or near completely for bar widths up to 50 μm. Near-complete adaptation transfer at this bar width requires that the receptive field of the adapting mechanism have a diameter of at least 100 μm—so that the mechanism always samples both the bright and dark bars (see *Figure 7B*). A receptive field of this size would contain at least 500 rods given the rod density of ~0.1/μm² (*Osterberg, 1935*). Bar widths of 100 μm and larger produced incomplete adaptation transfer, proving an upper bound on the extent of spatial integration.

The experiments of *Figures 7 and 8* indicate that gain control is mediated by a mechanism that has access to signals from hundreds of rods. This degree of spatial pooling is required to support the rapid kinetics described in *Figure 5* and is present only downstream of the synapse between rod bipolar cells and AII amacrine cells (*Figure 1*).

## Gain changes are mediated in rod bipolar pathway

To further constrain the possible sites of gain control, we looked for transfer of gain changes from rod- to cone-mediated signals (*Figure 9*). The rod bipolar pathway shares the cone bipolar cell and its synapse onto the ganglion cell with the pathways that convey signals transduced in cones to the ganglion cells (*Figure 1A*). Transfer of gain changes from the rod bipolar pathway to the cone pathway would indicate that gain is controlled in a circuit element shared between the two pathways—most likely the cone bipolar cell itself. Failure of gain changes to transfer would indicate that the mechanism is located at a circuit element providing input only to the rod bipolar pathway.

The spectral separation of the photopigments in rods and long-wavelength (L) cones and the considerably higher gain of rod-mediated signals allowed separation of rod- and cone-mediated inputs to a ganglion cell (*Figure 9A*; *Dunn et al., 2007*). Long-wavelength light stimuli produced a rapid L-cone–mediated response superimposed on a slower rod-mediated response (purple trace in *Figure 9A*). This reflects the ~600-fold greater sensitivity of the L-cones compared to the rods for long wavelength light and

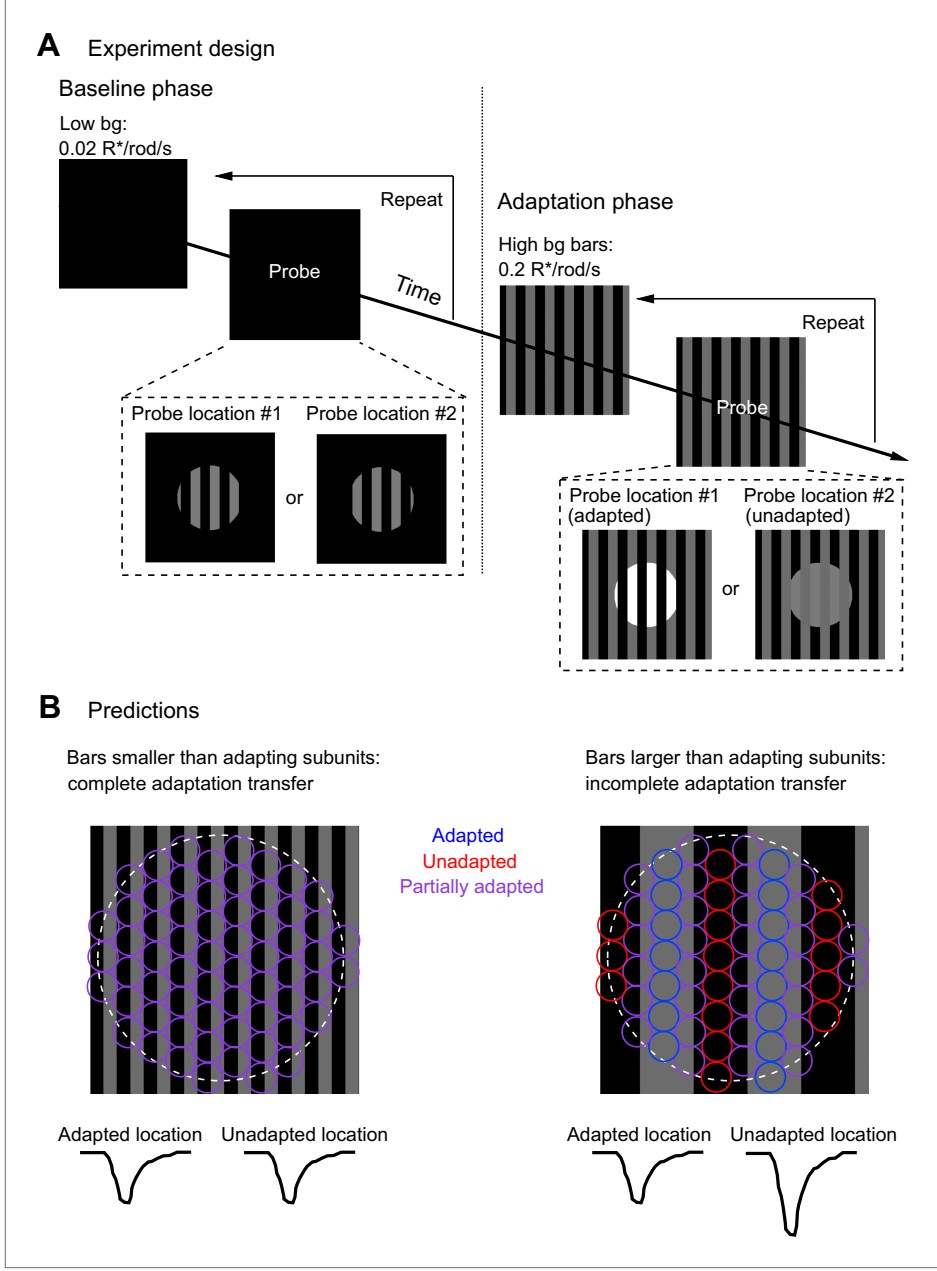

**Figure 7**. Measuring the spatial scale of gain control. (**A**) Diagram of the experimental design. The baseline phase consisted of at least 20 trials each of two probe patterns flashed on a uniform dim background (0.02 R*/rod/s). In the adaptation phase, the same two probe patterns were presented superimposed on an adapting background of bars of higher intensity (0.2 R*/rod/s), again for at least 20 trials each. One probe location was aligned with the adapting bars (adapted location), whereas the other was out of phase (unadapted location). (**B**) Repeating the experiment with a variety of adapting bar sizes tests the spatial scale of adaptation. Bars up to half the diameter of the adapting subregion of the receptive field would be expected to cause a complete transfer of adaptation from the adapted to the unadapted location. Bars wider than the diameter of the adapting subregion would be expected to cause incomplete adaptation transfer.

a roughly equal compensating factor for the higher gain of rod-mediated signals. Stimulation with dim short-wavelength light (<0.1 R*/rod and <0.03 R*/cone) generated a nearly pure rod-mediated response (blue trace in *Figure 9A*). Appropriate scaling of the strengths of the two flashes produced matched rod responses to long- and short-wavelength stimuli. Subtracting the rod-mediated response from the mixed response isolated the cone-mediated response (red trace in *Figure 9A*).

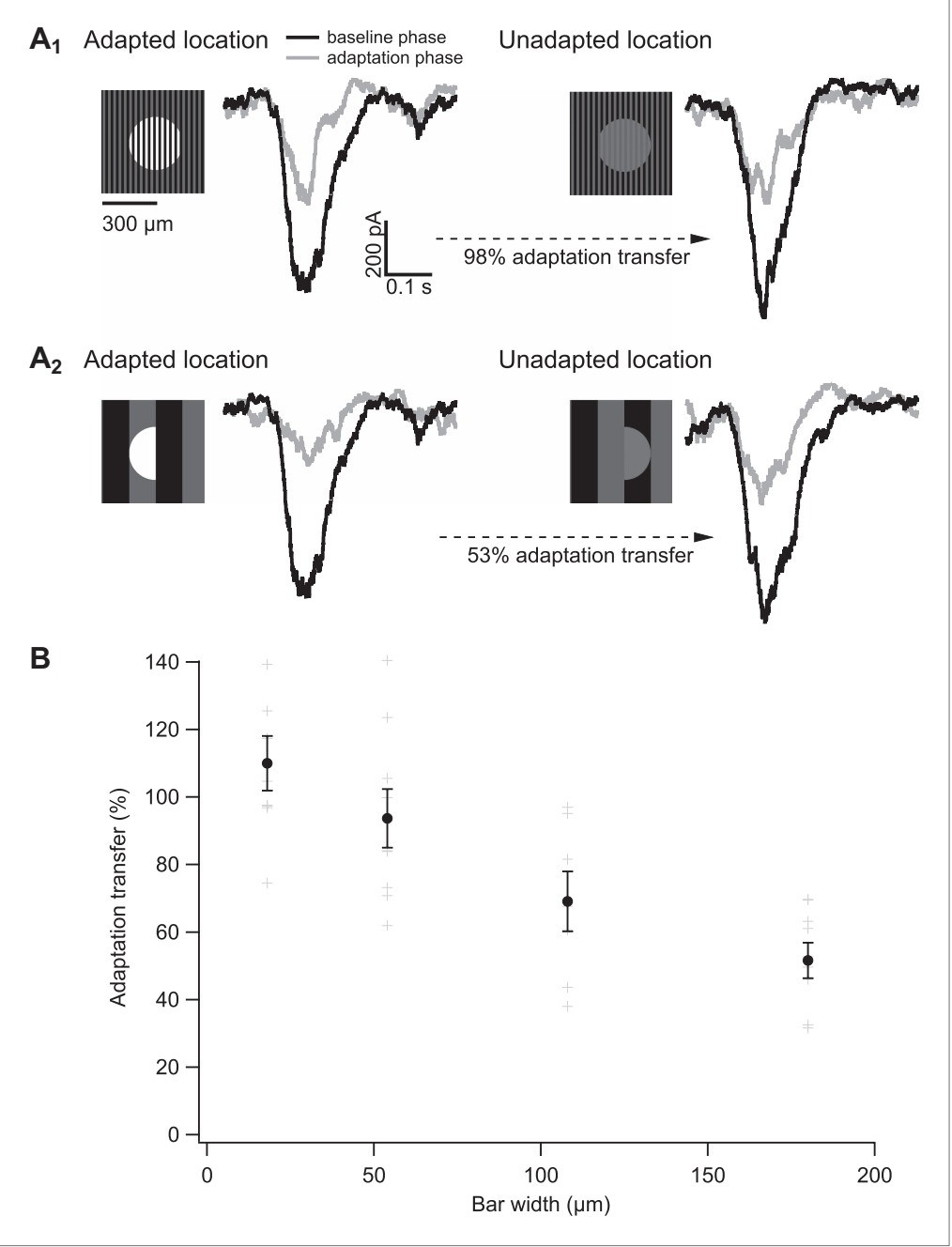

**Figure 8**. The spatial scale of gain control. (**A**) Responses of an example cell to probes during the baseline phase (black) and the adapting phase (gray) of the experiment at both the adapted and unadapted locations. Adapting bars were 18 μm wide in (**A₁**) and 180 μm wide in (**A₂**). (**B**) Adaptation transfer as a function of the width of the adapting stripes. Gray symbols are individual cells and black points are mean and SEM ($n$ = 7, 9, 7, and 8 for bar widths in ascending order).

We separated rod- and cone-mediated responses as in **Figure 9A** across a range of dim backgrounds (**Figure 9B,C**). Backgrounds that produced substantial reductions in gain of the rod-mediated signals failed to reduce the gain of the cone-mediated signals (**Figure 9B–E**). Thus, gain changes do not transfer from the rod pathway to the cone pathway, indicating that gain is controlled at a site prior to mixing of rod and cone signals in the cone bipolar cell. The lack of transfer from rod to cone pathways, together with the requirement that the gain control mechanism have access to signals from hundreds of rods (**Figure 8**), makes the AII amacrine cell and/or its gap junction with On cone bipolar cells likely sites.

## Discussion

Statistical fluctuations in the stream of photons incident on the retina make controlling the gain of rod-mediated signals at low backgrounds challenging. Yet, such gain control is important because it protects retinal signals from saturating as backgrounds increase while permitting the high amplification required to detect absorption of a few photons under fully dark-adapted conditions. Here, we have explored the properties of retinal gain controls operating near visual threshold. Below, we relate our findings to the dependence of human behavioral threshold on background and to the general speed/accuracy trade-off associated with detecting a change in the statistics of a noisy signal.

### Connection to behaviorally measured threshold

How do the properties of gain controls at low backgrounds in primate retina relate to human behavioral threshold verses intensity (TVI) measurements in humans? The region of the TVI curve in which threshold is unaffected by background light has long been associated with the intrinsic noise of the visual system (*Fechner, 1860*; *Barlow, 1956*). Threshold is increased only by backgrounds that produce a sufficient rate of photon absorption to exceed intrinsic noise. Human behavioral threshold begins to increase for backgrounds near 0.01 R*/rod/s, similar to the backgrounds (~0.02 R*/rod/s) at which primate ganglion cell thresholds began to increase. Both of these are considerably higher than the rate at which spontaneous activation of rhodopsin generates photon-like noise events in primate rods (0.003–0.004 R*/rod/s) (GD Field and F Rieke, unpublished). They are, however, consistent with measures of the total noise (i.e., both discrete and continuous) generated by macaque rods (*Schneeweis and Schnapf, 2000*).

The Rose–deVries region of the TVI curve has long been assumed to reflect an increase in noise without a reduction in gain (*Barlow, 1957*). Consistent with this view, the gain of retinal signals measured via electroretinograms begins to decline only at backgrounds ~fivefold higher than those that increase behavioral threshold (*Frishman et al., 1996*). Several other previous experiments, however, suggest that gain is reduced in the Rose–deVries range. First, behavioral brightness matching experiments, which should be sensitive to changes in neural gain but not noise, show a robust Rose–deVries region (*Brown and Rudd, 1998*). Second, the spike responses of toad ganglion cells exhibit an ~10-fold range of backgrounds over which the threshold required to reach a criterion spike rate increases in proportion to the square root of the background (*Donner et al., 1990*). Experiments in cat (*Barlow and Levick, 1969*; *Enroth-Cugell and Shapley, 1973*) and mouse (*Dunn et al., 2006*) also find adaptation at dim backgrounds, but with an exponent >0.5.

Our measurements of how gain control mechanisms affect both the signal and noise of the responses of On parasol ganglion cells provide a direct comparison to human psychophysics. The gain of signals in the rod pathway indeed decreased at low backgrounds, and noise increased much more slowly with increasing background than expected from Poisson fluctuations in photon absorption. The region of the ganglion cell TVI curve with slope 0.5 was limited to a narrow range of backgrounds. This suggests that post-retinal mechanisms contribute substantially to the more extended Rose–deVries region measured in behavioral studies.

### A panoply of gain control mechanisms

Our results place a number of constraints on the neural gain control mechanism that is engaged at the lowest backgrounds—for example, in starlight. The mechanism estimates the mean photon flux rather than its variance (*Figure 4*), it is rapid (*Figure 5*), it has access to signals from hundreds of rods (*Figure 8*), and it acts on rod-mediated but not cone-mediated signals (*Figure 9*). Previously described gain control mechanisms are sensitive to variance (reviewed by *Demb, 2008*), act at locations with insufficient rod convergence (*Dunn et al., 2006*; *Dunn and Rieke, 2008*), or affect cone signals (*Dunn et al., 2007*). The constraints above suggest a novel mechanism located in the AII amacrine cell or the synaptic connections it makes with cone bipolar cells (*Figure 5*). Recordings from these cells as well as the On cone bipolar cells to which they make gap junctions could uncover more mechanistic details.

Although we focused on gain changes in the inputs to On parasol ganglion cells, a location of the site of adaptation in the rod bipolar pathway indicates that it should be shared across ganglion cell types. Consistent with this prediction, On midget ganglion cells adapted very similarly to On parasol ganglion cells (data not shown).

The mechanism we identify here adds to a long list of gain controls in the retinal circuitry. Considering luminance alone, retinal ganglion cells encode stimuli over 10 log units of background, far exceeding

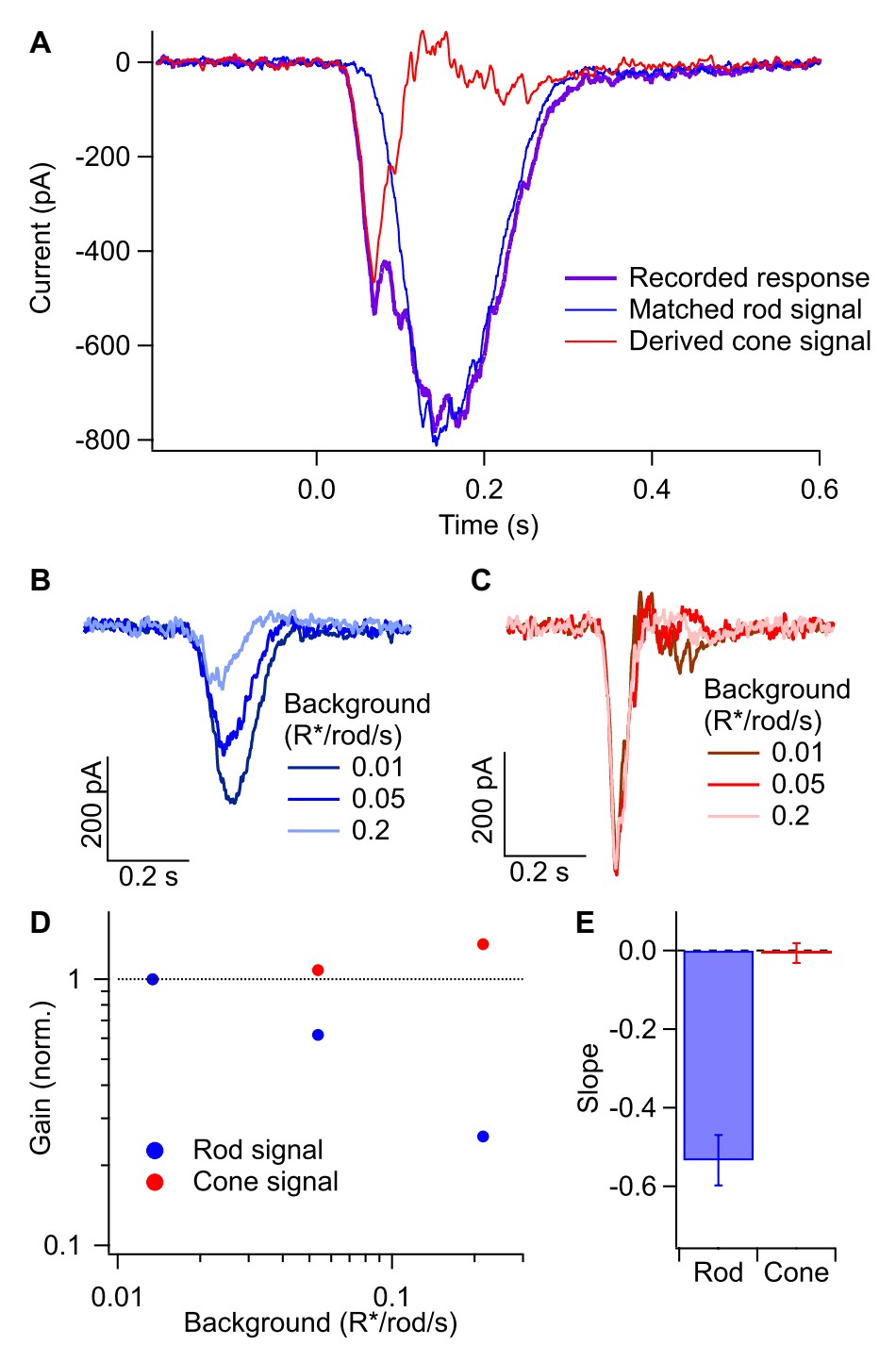

**Figure 9**. Gain control in rod and cone pathways. (**A**) Response to long-wavelength light (purple) and matched response to short-wavelength light (blue) were used to derive the isolated cone response (red). (**B**) Rod responses to the same flash intensity at three different backgrounds. (**C**) Cone responses recorded in the same cell as in (**B**). (**D**) Gain of the rod and cone responses for the cell in (**B** and **C**) plotted against the background on a log–log scale. Gain values have been normalized by the gain at the lowest background. (**E**) Slope of the background vs gain function on a log–log scale for rod and cone responses in the same population of cells. Error bars are SEM (*n* = 7).

the dynamic range of any single mechanism. In the rod pathway, there are at least three gain control mechanisms engaged across backgrounds, each matched to the rod convergence at the adapting location. Backgrounds producing approximately 10–20 R*/rod/s halve the gain of the phototransduction cascade in primate rods (*Schneeweis and Schnapf, 2000*). Backgrounds producing ~0.4 R*/rod/s halve the gain of the rod bipolar to AII synapse by reducing the number of releasable vesicles (*Dunn et al., 2006*; *Dunn and Rieke, 2008*; *Oesch and Diamond, 2011*). And, as shown here, backgrounds ~0.08 R*/rod/s engage a mechanism located downstream of the rod bipolar–AII synapse; the light levels at which this mechanism is engaged are consistent with backgrounds that are just detectable given the total noise generated by macaque rods (*Schneeweis and Schnapf, 2000*). Impressively, these mechanisms act in concert to create a smooth dependence of gain on background in the ganglion cell (e.g., *Figure 3A*), even extending to backgrounds at which signaling begins to switch from rods to cones and gain control mechanisms in the cone pathway become active (*Dunn et al., 2007*). At least five separate gain control mechanisms that contribute to a single smooth function relating background to gain over >5 log units on each axis (*Figure 3A*) suggest that this particular functional form for adaptation was a target of selective pressure in primate evolution.

## Adapting to a noisy signal

Adapting at low backgrounds is inherently dangerous as statistical fluctuations in photon absorption introduce substantial noise into the neural signals accessible to a gain control mechanism; such noise would be expected to cause gain to vary widely. The impact of noise in the signals controlling gain could be mitigated by not adapting until backgrounds increase enough to make more photons available, or by integrating over time and/or rods to reduce noise. Surprisingly, retinal gain controls appear to operate at the lowest backgrounds physically possible—that is, each absorbed photon within the pool of rods providing input to an adaptational mechanism lowers gain.

Adaptation at the rod bipolar to AII synapse approaches the sensitivity limit imposed by convergence and integration time (*Dunn and Rieke, 2008*; see *Figure 1B*). That is, a single absorbed photon reduces the gain of the next single photon response. Rod convergence onto AII amacrine cells is not known precisely in primate, but the onset of adaptation is consistent with a single photon absorption in the AII receptive field reducing gain. Assuming a 0.1 s integration time (*Figure 5*), a mechanism operating on a pool of 200 rods would have to reduce gain by 40% for each photon absorbed for the gain to be halved at 0.08 R*/rod/s; a pool of 800 rods could achieve this level of gain control with a 11% gain reduction per photon (see 'Modeling').

Vision in the dark requires high gain of the retinal circuitry conveying rod-mediated signals to ganglion cells. This high gain could easily lead to saturation of retinal signals as backgrounds increase. Indeed, the onset of retinal gain control coincides with the dimmest detectable backgrounds given the intrinsic noise of the rods. These considerations help explain why retinal gain controls are engaged by each absorbed photon, even when adaptation under these conditions is necessarily noisy.

## Materials and methods

### Recordings and light stimulation

All recordings were from On parasol ganglion cells in peripheral (eccentricity typically >30°; ~15° to 20° for the experiments of *Figure 9*) primate retina (*Macaca nemestrina*, *fascicularis*, and *mulatta*). Retina was obtained through the Tissue Distribution Program of the Regional Primate Center at the University of Washington. The retina was dark adapted for 1 hr while attached to the pigment epithelium and stored for up to 24 hr in a light-tight container at 32°C in Ames solution equilibrated with 95% $O_2$/5% $CO_2$. All subsequent procedures were performed under infrared light (>900 nm) using infrared-visible converters.

For recording, a roughly 4 × 4-mm piece of retina was removed from the pigment epithelium and mounted photoreceptor side down on a polylysine-coated glass coverslip forming the bottom of the recording chamber. The retina was secured by nylon wires stretched across a platinum ring. The retina was continuously superfused (8–10 ml/min) with warmed (32–34°C) Ames solution equilibrated with 95% $O_2$/5% $CO_2$. Calibrated light stimuli were delivered from LEDs focused on the retina by the microscope condenser. Spatial stimuli for *Figures 7 and 8* were delivered using

an OLED array (eMagin, Bellevue, WA). Stimulus intensities are given in terms of isomerizations per rod per second (R*/rod/s), based on the measured LED spectral output, the rod spectral sensitivity, and an assumed collecting area of 1 µm².

Spike responses were measured in the cell-attached configuration using pipettes filled with Ames solution. Whole-cell voltage clamp recordings were made using patch pipettes filled with 105 mM $CsCH_3SO_3$, 10 mM TEA-Cl, 20 mM HEPES, 10 mM EGTA, 5 mM Mg-ATP, 0.5 mM Tris-GTP, and 2 mM QX-314 (pH ~7.3 with CsOH, ~280 mOsm). Reported holding voltages have been corrected for the ~10 mV junction potential associated with this internal solution. Series resistance was typically 6–10 MΩ, and was compensated 50%.

All data came from parasol ganglion cells exhibiting high sensitivity when fully dark adapted. Recordings followed previous procedures (*Trong and Rieke, 2008*). We first used cell-attached recordings to test the spike response to a brief flash producing 0.001–0.002 R*/rod, and collected data only from cells that responded to these flashes with an average of 4–5 spikes. During experiments monitoring adaptation across a range of light levels (e.g., *Figures 2, 3 and 9*), we periodically returned to a 'reference' background near 0.05 R*/rod/s to check for slow changes in sensitivity. *Figures 10 and 11* illustrate the basic timeline of such experiments for cell-attached recordings of spike responses (*Figures 10*) and whole-cell recordings of excitatory synaptic inputs (*Figures 11*). *Figures 10B and 11B* show the changes in flash strength (gray line and right axes) used to generate near-equal responses

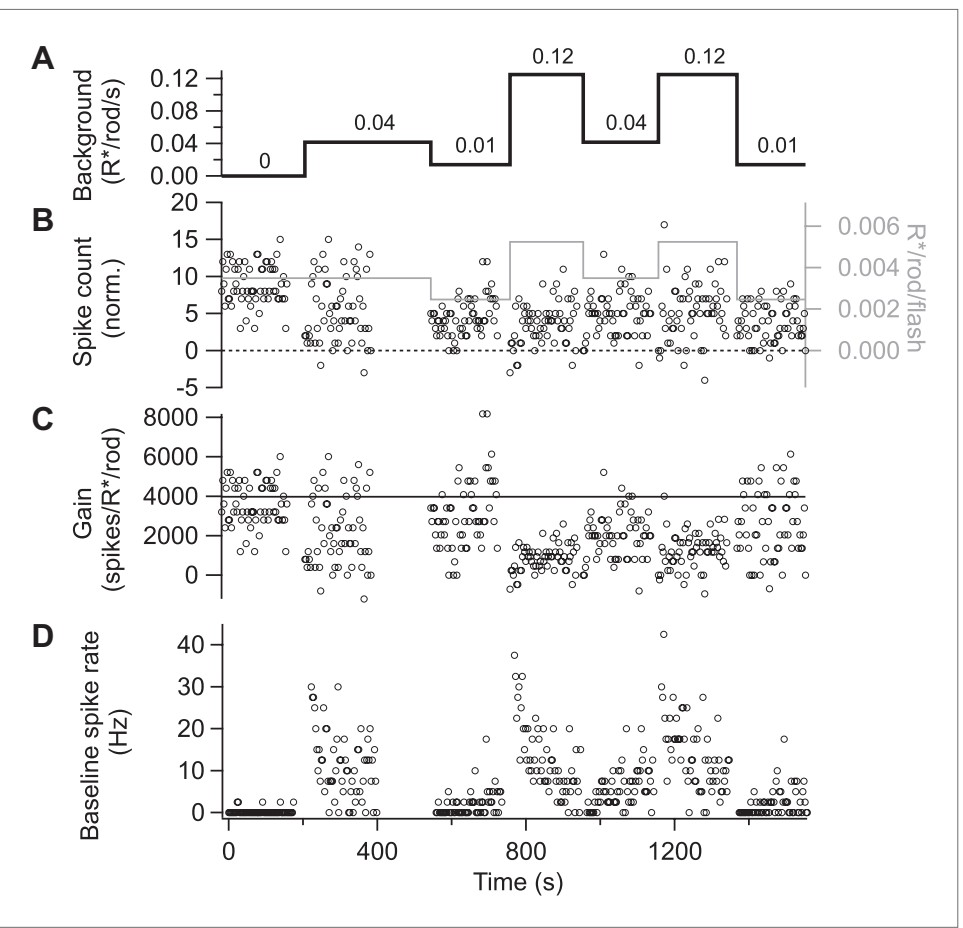

**Figure 10**. Gain control is stable and reproducible in spike responses. Spike responses from an example On parasol cell in primate retina. (**A**) Background light level. (**B**) Increase in spike count following a 10-ms flash (i.e., with the maintained spike rate before the flash subtracted). The test flash strength is indicated by the gray line and the right axis. Flashes were presented at 1-s intervals beginning 1 s after each background change. (**C**) Gain of the spike response in spikes per isomerization per rod (R*/rod). Gain was calculated by dividing the spike count in (**B**) by the flash strength. (**D**) Baseline spike rate measured prior to each flash.

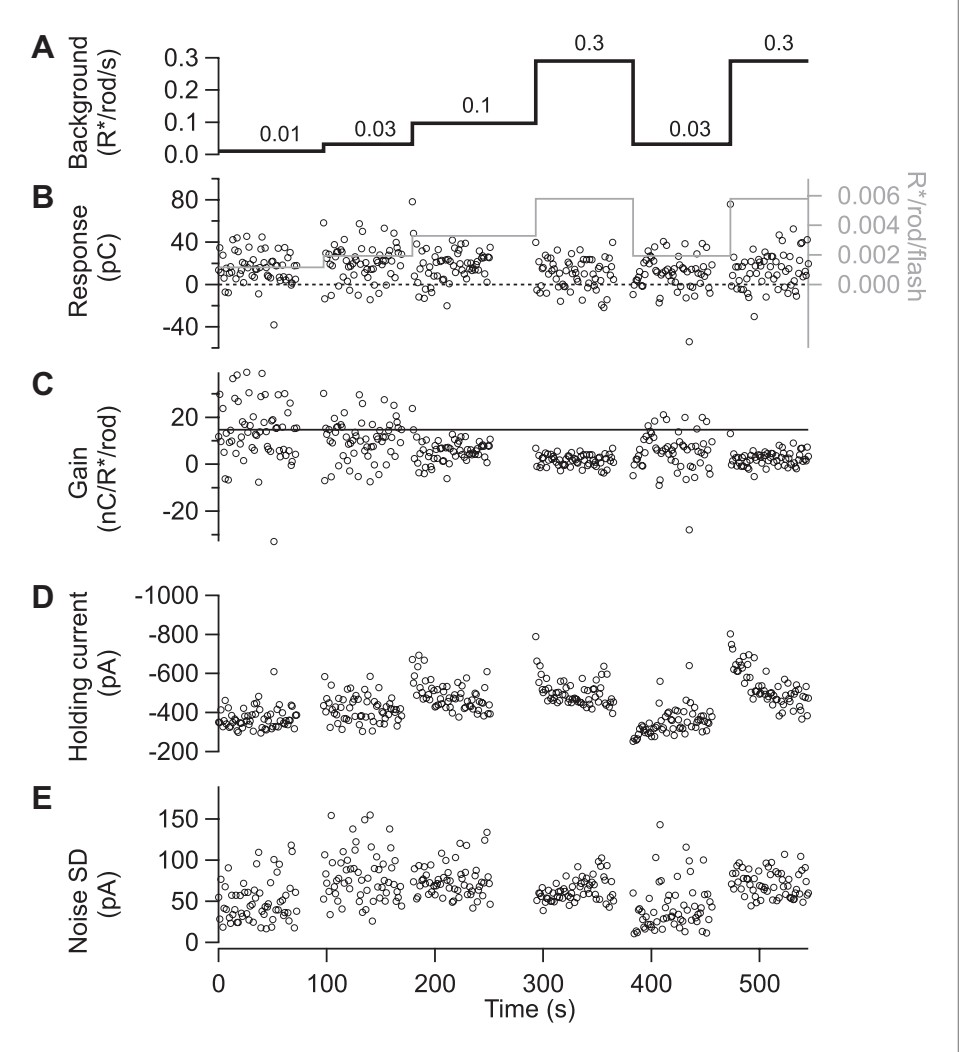

**Figure 11**. Gain control is stable and reproducible in excitatory input currents. (**A**)–(**D**) Excitatory synaptic currents following the same format as *Figure 10*. Data are from a different cell voltage clamped at −70 mV, the approximate reversal potential for inhibitory input. (**E**) Standard deviation of the current in the time interval preceding each flash.

across backgrounds. The figures also illustrate, based on raw data, some of the key features quantified in the 'Results': (1) the sensitivity of gain to weak backgrounds (*Figures 10A–C and 11A–C*); (2) the rapid changes in gain following changes in background (*Figure 10C and 11C*); (3) the slow changes in baseline firing rate (*Figure 10D*) and excitatory synaptic input (*Figure 11D*) following changes in background; and (4) the changes in noise following changes in background (*Figure 11E*).

## Analysis

A time window containing the flash response was chosen for each background and flash intensity by selecting the time points at which the average response exceeded 20% of its maximal value. The response amplitude was computed as the integrated current during this time window minus a baseline current recorded before the flash. Using a fixed time window across backgrounds and flash strengths yielded similar results. Gain values in *Figures 2, 3, 10, and 11* and *Figure 5—figure supplement 1* are the response amplitude divided by the flash strength.

We projected flash responses along a template to compute flash detection thresholds (*Figures 2 and 3*). The template was the average current trace in the time window described above. For each trial, we defined the signal $\gamma$ as the projection of the measured current along the template divided by the flash strength, and the noise $\nu$ as the projection of the current during a time window before the

flash. Detection threshold $\theta$ was defined as the flash strength required to give a signal-to-noise ratio of 1 according to the following equation:

$$\theta = \gamma / \sigma_v.$$

Gain and threshold of spike responses were computed analogously using the peristimulus time histogram to define the response time window for spike counts (*Figure 2*).

### Modeling

Probabilities of photon absorptions depending on background, integration time, and rod convergence (*Figure 6*) were computed from a Poisson distribution as follows. For a given background ($\mu$) in units of R*/rod/s, an integration time ($\tau$), and a rod convergence ($C$) per neural element, the probability of a neural element collecting a particular number ($k$) of photons follows Poisson statistics.

$$\lambda = \mu \tau C$$

$$P(k, \lambda) = \frac{\lambda^k e^{-\lambda}}{k!}.$$

*Figure 6* plots the combination of values of $\mu$, $\tau$, and $C$ required for half of the neural elements to collect at least one photon.

$$P(\lambda) = \sum_{k=1}^{\infty} \frac{\lambda^k e^{-\lambda}}{k!} \geq 0.5.$$

In the 'Discussion', we compute the gain reduction achieved for several size pools of converging rods. If each photon ($k$) collected by a neural element reduces the gain for the next photon response by a constant factor ($\alpha$), the gain as a function of $k$ and $\alpha$ can be computed as follows.

$$\gamma(k, \alpha) = (1 - \alpha)^k.$$

Therefore, the gain of a collection of neural elements ($\Gamma$) collecting photons according to Poisson statistics (normalized to $\Gamma = 1$ for maximal gain) is computed as follows.

$$\Gamma = \sum_{k=1}^{\infty} P(k, \lambda) \gamma(k, \alpha).$$

## Acknowledgements

We thank Petri Ala-Laurila, Juan Angueyra, Jeff Diamond, and Michael Rudd for helpful comments, and Mark Cafaro and Paul Newman for technical assistance.

## Additional information

### Funding

| Funder | Grant reference number | Author |
| --- | --- | --- |
| National Institutes of Health | EY11850 | Fred Rieke |
| Howard Hughes Medical Institute | | Fred Rieke |
| Helen Hay Whitney Foundation | | Gregory W Schwartz |

The funders had no role in study design, data collection and interpretation, or the decision to submit the work for publication.

## Author contributions

GWS, FR, Conception and design, Acquisition of data, Analysis and interpretation of data, Drafting or revising the article

## Ethics

Animal experimentation: Primate (*Macaca fascicularis, fascicularis* and *mulatta*) retinas were obtained through the Tissue Distribution Program of the Regional Primate Research Center at the University of Washington. All procedures were approved by the institutional animal care and use committee (protocol 4140-01).

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
