## [Decision Letter]

Thank you for choosing to send your work entitled “Controlling gain one photon at a time” for consideration at *eLife*. Your article has been evaluated by a Senior editor and 2 reviewers, one of whom is a member of our Board of Reviewing Editors.

The following individuals responsible for the peer review of your submission want to reveal their identity: Eve Marder, Senior editor; Ronald L. Calabrese, Reviewing editor; Marla Feller, peer reviewer.

The Reviewing editor and the other reviewer discussed their comments before we reached this decision, and the Reviewing editor has assembled the following comments to help you prepare a revised submission.

The authors describe electrophysiological experiments designed to determine if and how adaptation occurs at the level of ganglion cells at the lowest background light levels in macaque monkeys by recording spike output and EPSCs from parasol ganglion cells in isolated retinae. They show by systematic experiments that retinal adaptation begins at the dimmest detectable backgrounds given the intrinsic noise of the rods. Adaptation occurs rapidly at the level of AII Amacrine cells when as few as 1 or 2 photons are absorbed within a collection of several hundred rods. They successfully compare their experiments to human psychophysical experiments: human behavioral threshold begins to increase for backgrounds near 0.01 R*/rod/s, similar to the backgrounds (∼0.02 R*/rod/s) at which macaque ganglion cell thresholds began to increase.

The reviewers were enthusiastic about your work, found it exciting, and deemed its potential significance for the field quite high. Nonetheless, they felt that the manuscript itself did not do justice to the work, and that it made difficult reading. Therefore, as outlined below, it will require substantial editorial work before potential publication.

1) A general reader interested in how we see and adapt at low light levels could use a lot more guidance by the authors. For example, Figure 5A could be more colorful and used as a guide in the Introduction. We could be given a succinct summary of relevant retinal circuitry in Introduction. What is provided may not be enough especially without reference to a figure and some explanation of how the signals are transmitted (e.g., excitation inhibition, electrical coupling). Similarly the choice of the On parasol ganglion cell could be better rationalized at the beginning of Results.

2) The authors should provide clear definitions and use consistent language for describing “gain”, “noise”, and “background” throughout the manuscript.

* Background: the authors use “background”, “background lights”, or “stimulus” to describe changes in the background illumination. In the text describing Figure 6, this was particularly confusing, using terms like “test patterns”, “probe patterns”, and “adapting stimulus”. The authors should be explicit that changes in background illumination are what they are using for inducing adaptation.

* Gain: Is it really necessary to use so many measures of gain? Is the “signal gain” plotted in Figure 2A the same as the “gain” in Figure 4? Is the y-axis of Figure 4—figure supplement 1A the same as Figure 4—figure supplement 1B (where the units of the Gain are different, one using /flash and the other rods/sec). How does it differ from “gain factor” in Figure 4B? Why is gain sometimes computed as “Gain slope”? We realize that different definitions are more appropriate for comparing to different models, but it is tough on the reader. In addition, the authors need to make clear that the gain changes in the circuit affect both signal and noise.

3) The list of “qualitative features of adaptation” was particularly lacking in precise language. “Inhibitory synaptic inputs were small” (how small?), “…did not show obvious structure” (what is meant by “obvious structure”?). We would strongly recommend the author forgo this format for the text and give a clear, quantitative description of the results, and clear definitions of the quantities signal, noise, and gain for both cell attached and whole cell recordings.

---

## [Author Response]

We certainly want our work to be broadly accessible, and we appreciate the feedback that the previous version of the paper did not do that. As detailed below we have made major changes to the text and figures. This includes a new Figure 1, reordering several of the other figures, and rewriting substantial parts of the text, including the Introduction.

*1) A general reader interested in how we see and adapt at low light levels could use a lot more guidance by the authors. For example, Figure 5A could be more colorful and used as a guide in the Introduction. We could be given a succinct summary of relevant retinal circuitry in Introduction. What is provided may not be enough especially without reference to a figure and some explanation of how the signals are transmitted (e.g., excitation inhibition, electrical coupling)*.

Thank you for the suggestion. We have rewritten the Introduction to be more specific about the circuitry operating under the conditions of our experiments, and what is known from past work about gain control mechanisms operating in this circuitry. As part of that revision we created a new Figure 1 with a more detailed diagram of the retinal circuitry. The diagram and caption include details about the type of synapses and the rod convergence at each location, as well as the location of previously described sites of gain control. We feel that this figure, along with supporting changes in the Introduction, provides a nice graphical introduction to the retinal circuitry and the problem of gain control.

*Similarly the choice of the On parasol ganglion cell could be better rationalized at the beginning of Results*.

We revised this paragraph to emphasize that the On parasol cells provide a good readout of activity in the rod bipolar pathway. We now specifically make the point that this pathway is shared across ganglion cell types at low light levels, and hence gain control mechanisms located in shared components of the circuit should be conserved across ganglion cell types (as we observed in Midget ganglion cells, as noted in the Discussion). We also now anticipate in the text that the gain control mechanism we identify and characterize is indeed located at a shared location in the rod bipolar pathway. We believe this revised paragraph does a better job at emphasizing the key reasons that we chose to study On parasol cells.

*2) The authors should provide clear definitions and use consistent language for describing “gain”, “noise”, and “background” throughout the manuscript*.

** Background: the authors use “background”, “background lights”, or “stimulus” to describe changes in the background illumination*.

Indeed in the original paper we used these terms inconsistently. We have fixed this in the revised version. “Background” is now used consistently and terms such as “mean light level”, “mean luminance”, and “mean stimulus” have been removed in favor of “background”. Similarly, we have consistently used “gain control” in place of adaptation, since it more directly relates to the measures of response gain shown throughout the results.

*In the text describing Figure 6, this was particularly confusing, using terms like “test patterns”, “probe patterns”, and “adapting stimulus”. The authors should be explicit that changes in background illumination are what they are using for inducing adaptation*.

Figure 7 (previous Figure 6) has been substantially revised (new panels A and B) along with the text describing the logic behind the experiment. Terminology in the text now directly references elements of Figure 7A and B. We have also made the terminology used to describe this experiment more consistent with the rest of the Results section. We hope these changes clarify the logic behind the experiment and interpretation of the results.

** Gain: Is it really necessary to use so many measures of gain? Is the “signal gain” plotted in Figure 2A the same as the “gain” in Figure 4? Is the y-axis of Figure 4—figure supplement 1A the same as Figure 4—figure supplement 1B (where the units of the Gain are different, one using /flash and the other rods/sec). How does it differ from “gain factor” in Figure 4B? Why is gain sometimes computed as “Gain slope”? We realize that different definitions are more appropriate for comparing to different models, but it is tough on the reader. In addition, the authors need to make clear that the gain changes in the circuit affect both signal and noise*.

Thanks very much for this suggestion. Many of the inconsistencies were oversights on our part rather than different analyses or measures. All figures now use either gain in appropriate units (nC/R*/rod for excitatory input currents or spikes/R*/rod for spike signals), or a normalized value, “gain (norm.),” if comparing across conditions where relative gain values are more appropriate than absolute values. Gain is clearly defined when it first comes up in the Results. We have replaced “gain slope” with “slope,” and we explain that this slope on a log-log plot is equivalent to “the exponent describing the dependence of gain on background” in the third paragraph of the section, “Dependence of gain and noise on background”. Also, the term “signal gain” has been removed as it was indeed confusing since signal and noise are both affected.

*3) The list of “qualitative features of adaptation” was particularly lacking in precise language. “Inhibitory synaptic inputs were small” (how small?), “…did not show obvious structure” (what is meant by “obvious structure”?). We would strongly recommend the author forgo this format for the text and give a clear, quantitative description of the results, and clear definitions of the quantities signal, noise, and gain for both cell attached and whole cell recordings*.

Obviously our original approach to introducing the results did not work well. As suggested, we removed the “qualitative features” section and replaced it with a quantitative description of the results. In making this change, we moved Figure 1 from the previous draft into the Methods section and replaced it with a simpler figure (new Figure 2) showing raw data for excitatory input currents and spikes in the same cell, and quantifying the similarity of the spike response and excitatory input. This enabled us to introduce “noise” with a precise definition in the next figure (Figure 3) after an initial characterization of the change in gain.